# Bringing legislation to life: Navigating intersectional barriers in implementing the Sindh Empowerment of Persons with Disabilities Act (2018) for the healthcare of children with disabilities

Muhammad Asim[1]*, Waqas Hameed[1], Sara Rotenberg[2], Dalia Chowdhury[3], Abid Lashari[4], Munazza Gillani[5], Abdul Ghaffar[1‡], Hannah Kuper[2‡]

1 Department of Community Health Sciences, Aga Khan University Karachi, Karachi, Pakistan,
2 International Centre for Evidence in Disability, London School of Hygiene & Tropical Medicine, London, United Kingdom, 3 Department of Rehabilitation and Health Services, University of North Texas, Denton, Texas, United States of America, 4 National Disability and Development Forum, Sindh, Pakistan,
5 Sightsavers International, Islamabad, Pakistan

‡ Joint last authorship.
* asim.muhammad@aku.edu

## Abstract

The Sindh Empowerment of Persons with Disabilities Act 2018 (SEPD Act) mandates inclusive health services for persons with disabilities. This study examines the challenges faced in implementing the disability Act's health-related provisions, with a focus on children with disabilities in Sindh, Pakistan. A qualitative exploratory study was conducted in four districts of Sindh province. A total of 40 key informant interviews were undertaken with government officials, health administrators, pediatricians, and disability activists using a purposive sampling technique. The data were analyzed through inductive and deductive thematic analysis using the Missing Billion disability-inclusive health system framework. We identified several critical systemic gaps in the implementation of the SEPD Act in Sindh province. Weak governance, including limited intersectoral coordination, frequent bureaucratic turnover, and lack of private-sector engagement impedes implementation progress. The absence of sanctioned budget for inclusive health initiatives, particularly the lack of structured health insurance programs and limited availability of assistive devices, restrict healthcare utilization for children with disabilities and forces families to incur substantial out-of-pocket expenses for rehabilitation and healthcare. Effective delivery of health services are constrained by inadequate disability-inclusive infrastructure, insufficiently trained healthcare providers, and lack of newborn screening programs for early detection and timely intervention. Furthermore, lack of systematic collection and reporting of disability-specific data within the Health Information System limits evidence-informed policy and decision-making and the development of effective referral mechanisms for diagnosis, treatment, and rehabilitation of

**Data availability statement:** The qualitative interview transcripts contain potentially identifiable information from a small number of participants, including officials from the Department of Empowerment of Persons with Disabilities (DEPD), disability advocates, and physicians (genetic pediatric specialists). For ethical and confidentiality reasons, the full transcripts cannot be made publicly available. De-identified data underlying the findings reported in this article may be made available upon request to the Principal Investigator, Dr. Muhammad Asim (asim.muhammad@aku.edu). Requests may also be directed to the Chair Ethics Committee, Aga Khan University, Karachi (erc.pakistan@aku.edu).

**Funding:** This work was supported by National Institute of Health (Health Research Institute), Islamabad, Pakistan. This grant was awarded to MA and Grant Reference Number RIG-22/R2-111/RDC/AKU. The funder had no role in study design, data collection and analysis, decision to publish, or preparation of the manuscript.

**Competing interests:** The authors have declared that no competing interests exist.

children with disabilities. Systemic weaknesses impede the delivery of disability-inclusive healthcare in Sindh, despite the presence of supportive legal frameworks. Our findings recommend strengthening the implementation of the SEPD Act through strategic planning, coordination, and priority setting at both provincial and district levels. Furthermore, it is essential to integrate strategies for a disability-inclusive health system at all administrative levels to ensure that the needs of persons with disabilities are prioritized.

## 1 Background

Globally, more than one billion people, approximately 16% of the world's population, are disabled, equating to one in six individuals [1]. A recent UNICEF report estimates that this figure includes nearly 240 million children who have moderate or severe disabilities, representing 1 in 10 children worldwide [2]. The UN Convention on the Rights of Persons with Disabilities (UNCRPD) defines disability as a long-term physical, mental, intellectual, or sensory impairment that, in interaction with environmental and societal barriers, limits equal participation in society [3].

Children with disabilities remain among the most marginalized groups worldwide, facing significantly worse health outcomes [2]. Evidence indicates that they have greater unmet healthcare needs, incur higher catastrophic health expenditures, and require additional medical support compared to children without disabilities [4,5]. For example, children with disabilities experience nearly twice the burden of acute respiratory infections (ARI), diarrhea, fever, and malnutrition globally [2]. Furthermore, they and their caregivers encounter multiple barriers to healthcare access, including lack of awareness, limited autonomy due to extreme poverty (7–10), transportation issues, denial of care, substandard treatment, social stigma, and inadequate inclusive health infrastructure [6–9].

Over the past few decades, global advocacy for the rights and inclusion of people with disabilities for disability-inclusive health systems has gained significant attention [4,5,10]. In 2006, the UNCRPD emphasized the right to healthcare for people with disabilities and the importance of delivering the highest standard of healthcare to individuals with disabilities, ensuring the elimination of discrimination [3]. Similarly, Sustainable Development Goal 3 (SDG-3) aims to ensure 'healthy lives and well-being for all at all ages' across the globe, 'leaving no one behind', which implicitly includes people with disabilities [6].

Despite these international commitments, delivering high-quality health services to all without causing financial hardship remains challenging, though it is essential to achieving Universal Health Coverage (UHC) [7]. These complex barriers result in worse health outcomes for children with disabilities, who are among the most vulnerable groups in the population [8,9,11,12]. In low- and middle-income countries (LMICs), children with disabilities are particularly neglected in health policy and planning, reinforcing their cycle of unique and unmet healthcare needs [13–15].

Inclusive healthcare and rehabilitation services for children with disabilities remains one of Pakistan's most neglected public health challenges [8]. The Employment and Rehabilitation Ordinance of 1981 was the first legislative initiative related to disability, introduced over 30 years after the country's independence [16,17]. After this ordinance, various laws and policies have been developed at both national and provincial levels to empower people with disabilities [16–18]. However, most of these laws and policies initiatives are widely neglected prioritizing healthcare domains or children with disabilities. As a result, disability-inclusive health-related initiatives remain widely neglected at both national and provincial levels [8,19]. The implementation of health-related provision of the existing disability laws continues to face significant challenges in Pakistan.

The 18th Amendment to the Constitution in 2010 enhanced provincial autonomy by transferring powers from the federal government to provinces [20]. This reform granted provinces greater legislative and financial autonomy to prioritize key initiatives based on local needs. The Sindh Empowerment of Persons with Disabilities (SEPD) Act 2018 is the first comprehensive legal provision in Sindh province that endorses equity and empowers persons with disabilities [18,21]. This act ensures barrier-free access to health and rehabilitation services from both the public and private sectors, provision of assistive technology (AT) devices, affordable health insurance, private sector engagement for free health services, newborn screening and community engagement for disability awareness (see Box 1). Under the act, children with disabilities are able to obtain disability certificate, linked to the child's registration certificate based on medical and functional assessments. This certificate helps with access to health, rehabilitation and social protection services. However, there is limited evidence on how effectively its health-related provisions have been implemented with the focus on children with disabilities. Global evidence indicates that children with disabilities face higher unmet healthcare needs and are more vulnerable to disease.

The objective of this study is to investigate the challenges in implementing the health-related provisions of the Sindh Disability Act 2018, in Sindh province. In particular, this study examines whether child-centered legal frameworks in Sindh province are effectively implemented in practice.

## Box 1. The Department of Empowerment of Persons with Disability Act 2018 mandate to health and rehabilitation services by Sindh government

- Provision of barrier-free access to public health services and infrastructure without any discrimination or cost.

- Provisions of aids and appliances, diagnostic services, medicine, and corrective surgeries are free of charge.

- Provision of affordable and subsidized health insurance through private insurers and state-owned insurance entities to address discrimination.

- Private sector health service providers' engagement to provide quality service at affordable rates.

- Implementation of the Sindh newborn screening act 2013 to build capacity of health facilities for early identification and timely intervention to manage disabilities.

- Engagement of the medical and scientific communities to scientifically identify the causes of disabilities and create public awareness to prevent disabilities.

## 2 Methodology

### 2.1 Ethical statement

This study was approved by the Aga Khan University Ethical Review Committee (ID: ERC.6808- 21404). Written informed consent was taken before starting the interview. There was a volunteer participation of study participants in this study. This

study is the part of a project "Situational Analysis of healthcare needs of children with disabilities in Sindh" that was funded by Health Research Institute Islamabad, Pakistan.

## 2.2 Study design and settings

A qualitative exploratory research design was employed to understand the implementation challenges of the health-related clauses of the Sindh Disability Act 2018. The data were collected from a diverse stakeholder through key-informant (in-depth) interviews. This study was conducted in four purposively selected districts of Sindh province: Karachi, Hyderabad, Sukkur, and Dadu. These districts were purposively selected for geographical diversity of the province. These districts are located in southern (Karachi), central (Hyderabad and Dadu), and northern (Sukkur) regions of Sindh province. This selection of districts reflects a diverse health system context in relation to health services, ranging from metropolitan cities to semi-urban and rural districts. The Department of Empowerment of Persons with Disabilities (DEPD) has divisional and district offices in all these selected districts for our study [18,22].

## 2.3 Study participants and sample size

Four different types of stakeholders were interviewed for this study: (1) bureaucrats and policymakers from the department of empowerment of persons with disabilities, (2) district health administrators, (3) disability activists (non-government), and (4) pediatricians from district and tertiary care hospitals. Table 1 presents the detailed sample breakdown by category. The interviews were collected until the data saturation was achieved. The final sample size of 40 participants was agreed when the research team did not capture new information or themes after 35 interviews. An additional five interviews were conducted to confirm saturation after the pilot analysis. Written informed consent was obtained from all study participants.

## 2.4 Interview guide development

Three semi-structured interview guides were developed for government officials, non-government officials, and physicians. The guides were designed based on the Missing Billion Health System Framework (S1 Text). The major questions and prompts included in the interview guide were related to affordability of health services, autonomy and awareness, health system factors including government and leadership, health financing and data and evidence on disability-inclusive health system framework. The interview guides were pilot tested with five participants in Karachi. The testing allowed for refinement of the wording of the questions and the order in which they are asked, ensuring that the content is culturally sensitive to the target audience and relevant to their community. Minor adjustments were made to the wording of the probe questions to clarify the follow-up questions.

## 2.5 Data collection

The data collection took place between 01 January 2023 and 30 May 2023 by a team consisting of two experienced social scientists (one male, one female). Interviews with key informants were conducted in Urdu/Sindhi at places preferred by

Table 1. Sample size and distribution of study participants (n = 40).

| Types of participants | Number of interviews | Percentage |
|---|---|---|
| Leaders and policymakers (Government Officials) | 8 | 20.0 |
| Health administrators (Government Officials) | 12 | 30.0 |
| Disability activists (Non-Government Officials) | 10 | 25.0 |
| Pediatricians | 10 | 25.0 |

the participants for their convenience. The participants consented to the interviews, which were recorded with an audio device and transcribed verbatim for accuracy. The field notes and informal discussion with study participants provided contextual and rich information to further probe the central study questions. A debriefing session was held after each interview to contextualize the participant's answers. Each interview lasted between 30 and 45 minutes.

## 2.6 Data analysis

The data were analysed by combination of deductive-inductive thematic approach. Analysis followed five steps: familiarization, initial coding, theme development and refinement, and interpretation [23]. First, eight transcripts representing diverse participants were independently reviewed to conduct open coding and generate an initial set of codes. These codes were consolidated into a preliminary codebook through discussion among the co-authors. Next, the preliminary codes and emerging themes were mapped inductively onto the domains of the Missing Billion Disability-Inclusive Health System Framework [24] to structure interpretation and cross-case comparison. Microsoft Excel was used for qualitative data management, coding, and development of an analysis matrix, which was appropriate given the sample size enabled transparent team-based comparison and auditability of coding decisions when the research team don't have access to computer-based data analysis software. To enhance analytical rigor, two researchers double coded the 5 randomly selected transcripts and compared coding outputs; discrepancies were discussed with a senior investigator until consensus was reached. The intercoder reliability was 87%. The final codebook and framework-mapped matrix were then used to organize and report the final themes.

## 2.7 Reflexive statement

This research team comprised of mixed specializations including academicians (MA, WH, HK, SR, DC, AG), professionals from the development sector (AL, MG), and specialists in health policy and systems (MA, WH, HK, SR, DC), with expertise in disability and universal health coverage. The first author (MA), based in Sindh and without a disability, conducted interviews with the support of a research assistant who coordinated with all relevant stakeholders and scheduled appointments as per their convenience. This geographic proximity may have introduced certain positionality bias, such as familiarity, into the findings. During the data analysis, we reflected on how our affiliation with a medical university, and a private hospital, may have influenced interactions with policymakers, administrators, activists, and providers. These potential influences were critically examined throughout the analytical process to minimize bias. To further address positionality and uphold rigor in data interpretation, the research team employed reflective journaling throughout the research process.

A co-author (AL), a person with a physical disability, was involved in the development of the DEPD legislation in Sindh province and served on the advisory committee of this Act. The authors are neither healthcare providers nor engaged in service delivery. The data coders faced some challenges in coding and analyzing the data specifically on children with disabilities, and some of the discussion was broadly focused on people with disabilities. To address this issue, a third data coder was engaged to finalize the coding where discrepancies arose between two data coders.

## 3 Results

Table 2 displays the socio-demographic characteristics of participants interviewed for this study. Most of the participants were male (75%), and just over half (53%) were in the 35–50 years age group. More than two-thirds of the participants were working in urban areas, and the average work experience of the participants was 19 years.

The qualitative findings have been presented using Missing Billion Disability-Inclusive Health System Framework [24] illustrated in Fig 1. We presented the findings into two broad themes from the framework: health system and service delivery domains. The questions related to demand-side factors were not included for stakeholders. This study focuses on children with disabilities, stakeholders frequently articulated challenges at the broader system-wide disability level. However,

**Table 2. Socio-demographic characteristics of the participants (n = 40).**

| Demographic characteristics | Categories | Frequency | Percentage |
|---|---|---|---|
| **Gender** | Male | 30 | 75% |
| | Female | 10 | 25% |
| **Age in years** | Up-to 35 | 8 | 20% |
| | 36-50 | 21 | 53% |
| | 51 and more | 11 | 28% |
| **Duty station** | Rural | 13 | 33% |
| | Urban | 27 | 68% |
| **Working experience in years: Mean (±SD)** | | 19(±6) | |

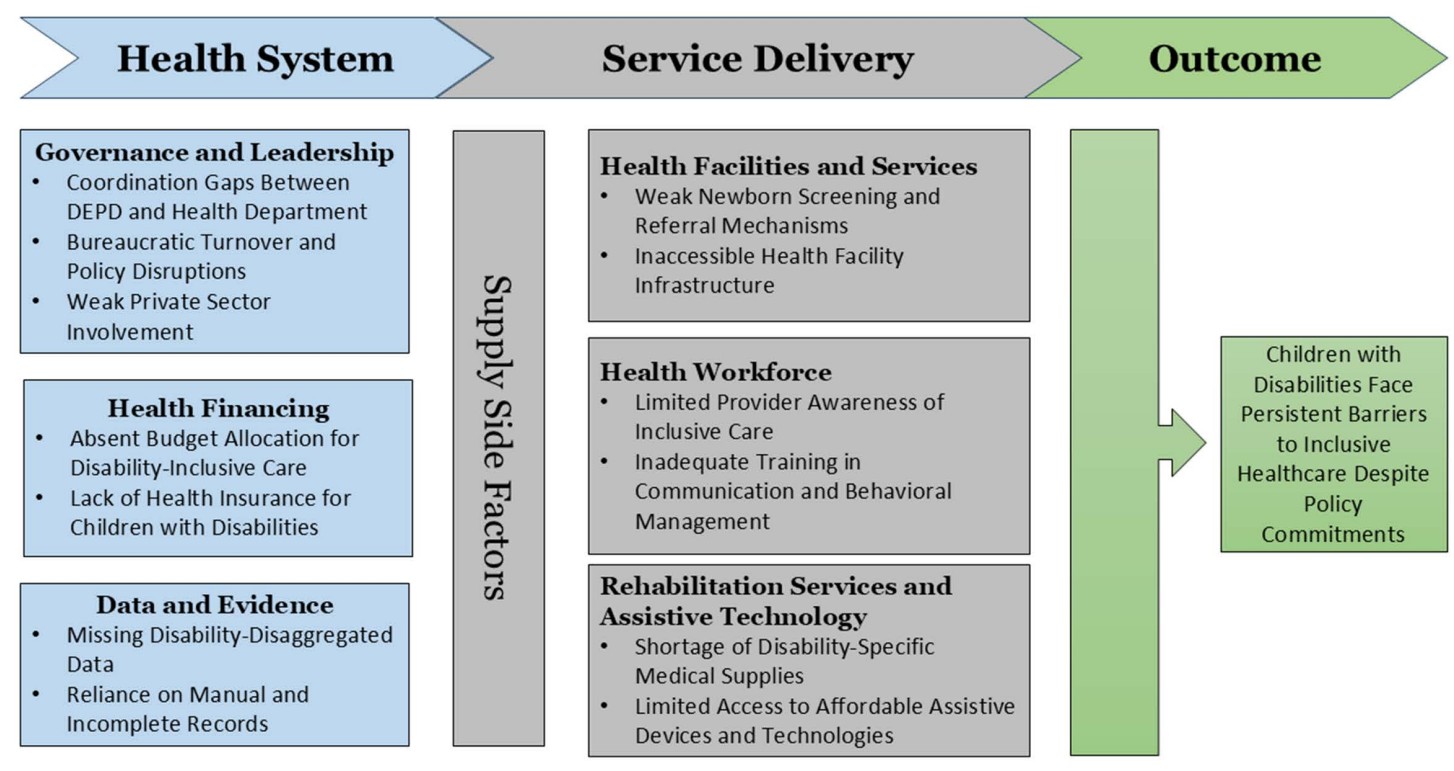

**Fig 1. Summary of findings using the adapted Missing Billion Health System Framework.**

we foreground child-relevant data derived from these system-level narratives, recognizing children with disabilities as a sub-group disproportionately affected by the absence of disability-inclusive health systems.

### 3.1 Health system level factors

Our interviews identified health system–level factors that influence the provision of disability-inclusive healthcare to children with disabilities, highlighting three interconnected sub-themes: (i) leadership and governance, (ii) health financing and insurance, and (iii) health information systems.

### 3.2 Governance and leadership: Implications for child healthcare

Our interviews highlighted that governance and leadership challenges directly shape how and whether children with disabilities are identified, referred, and supported within the health system.

**3.2.1 Coordination gaps between DEPD and health department.** The governance and leadership of the Department of Empowerment of Persons with Disabilities (DEPD) evolved from the prior Special Education Department, which was an independent entity before the enactment of the SEPD Act in the province. The special education department became the custodian of DEPD and prioritize special education, instead of disability-inclusive health in the province. Health is an independent provincial department.

Effective implementation of the Sindh Disability Act 2018 requires strong intersectoral collaboration between Department of Empowerment of Persons with Disabilities (DEPD) and health departments. The DEPD is a provincial government department responsible for implementing legal frameworks, policies and programs related to the rights, inclusion, and welfare of people with disabilities in Sindh. However, the health department is currently responsible for delivery of health services in the province. Our interviews reveal ongoing coordination gaps between DEPD and the health department, leading to inadequate collaboration and a lack of focus on prioritizing and implementing inclusive health initiatives. A government official highlighted these limitations:

*"We cannot make health policies and provide care... Health is an independent department. We can only do collaboration and coordination with other departments. Implementation of the health policies is the responsibility of the health departments."* (Government Official, Karachi)

Similarly, an NGO worker stressed the absence of structured mechanisms for collaboration:

*"There is no real collaboration between the DEPD and the Health Department to implement inclusive health services. Each department works separately. There should be a designated focal person in every department to lead efforts and provide updates to DEPD."* (Disability Activist, Hyderabad)

A health department administrator highlighted systemic exclusion, noting the lack of formal engagement in disability-related initiatives:

*"We were never called in the meetings [inclusive health], and we also never receive any letter related to [inclusive] health services for people and children with disabilities. We can equip our system if we have been informed through official correspondence of such [inclusive] initiatives."* (Health Department, Sukkur)

Participants highlighted that lack of intersectoral coordination widely affects children with disabilities, such as early screening, referral, rehabilitation, and continuity of care depend on structured collaboration between disability, education, and health sectors that are currently absent in the province.

**3.2.2 High turnover and policy disruptions.** Frequent transfers of government officials have been a serious hinderance in implementation of the Act and delivery of health services. Regular changes in leadership lead to discontinuity and reduce institutional memory, complicating efforts to incorporate disability-inclusive priorities into the health sector system. A government official pointed out this ongoing systemic issue.

*"The frequent transfer of bureaucrats and government officials disrupts policy implementation. It takes time for new officials to understand the scope of their work and establish collaboration with other line departments. These challenges delay the execution of disability-inclusive policies particularly for children."*(Government Official, Karachi)

Participants consistently noted that each leadership change necessitated a period of adjustment as new officials needed time to understand the Act's mandate and its importance for child healthcare. However, before significant progress could occur, many were replaced or transferred, resulting in a cycle of stalled implementation. As one respondent explained, officials often defaulted to existing agendas rather than introducing disability-inclusive initiatives:

*"Every leadership has its own priorities. Once someone gets an idea of this new disability-inclusive act mandate, they are transferred or replaced. Flagship initiatives are usually prioritized, and no new initiatives are taken by officials. (Disability Activist, Karachi)"*

For children with disabilities, high turnover of officials is particularly harmful, because of disruption and delays in continuity of child specific services, such as early intervention services, and child inclusive care pathways.

**3.2.3 Weak private sector involvement.** The Sindh Disability Act 2018 mandates private sector involvement in disability-inclusive services, but actual implementation is limited. Despite these legal requirements, private hospitals and clinics show low awareness and participation due to poor incentives, awareness, and lack of commitment, especially when majority of population, for different reasons, don't go to the public health facilities. As one government official acknowledged:

*"We have been working on it to engage the private sector to provide free healthcare services to disabled children and others… However, various proposals are taken into consideration, and it may take some time to materialize them." (Government Official, Hyderabad)*

A physician from a tertiary private hospital further highlighted information gaps:

*"We do not provide free healthcare services to children with disabilities because the government did not share such information and SOPs with us. We have collaborations with some NGOs that sponsor the cost of consultancy for children with intellectual disabilities." (Healthcare provider, Karachi)*

A disability activist explained the consequences of such systemic gaps

*"Many private hospitals are approachable and provide specialized care. The caregivers of such children can't seek treatment from private hospitals because they are too expensive and don't provide free services to anyone." (Disability Activist, Dadu)*

The governance and leadership challenges limit the translation of the Sindh Disability Act into practical, child-centred healthcare actions within routine health services.

## 3.3 Health financing

The interviewees reported that health financing constraints disproportionately affect children with disabilities, because of the current structures and allocation of health budget, especially in public sector, where there are no budgets for rehabilitation services, assistive devices, and long-term care.

**3.3.1 Lack of budget allocation for disability-inclusive care.** Dedicated financial investment is crucial for implementing disability-inclusive healthcare, but the government has not allocated funds for infrastructure, equipment, or workforce training. The authority responsible for developing opportunities for people with disabilities, DEPD, lacks the

necessary funds to carry out mandated initiatives and perceives inclusion as expensive. As reflected by a healthcare provider:

*"Disability-friendly health facilities require budget allocation to improve infrastructure, hospital equipment, and health-care provider training. Focusing on children [with disabilities] requires even more resources due to their unique needs." (Healthcare provider, Karachi)*

A government official also confirmed the lack of designated health-sector funding to DEPD. The health-related budget is directly transferred to district health authorities instead of DEPD

*"We allocate a separate budget for special education. Special children also receive some free medicine as other children receive from public hospitals. There is no specific budget allocation for it [disability]. I never heard about budget allocation to DEPD to take initiatives for overall and children with disabilities." (Government Official, Karachi).*

**3.3.2 Lack of health insurance for children with disabilities.** Although the Sindh Disability Act 2018 mandate provision of health insurance to people with disabilities, including children. However, there is no established structured insurance program in the province. The lack of comprehensive financing mechanisms exacerbates existing inequalities. A worker from an NGO stressed:

*"The health insurance system is not launched in our province. Free health insurance must be given to children. These children are more vulnerable and experience multiple diseases. The healthcare expenses are higher for children with disabilities as compared to other children." (Disability Activist, Dadu)*

A physician similarly noted the reliance on families' own resources:

*"I do not know about free health insurance for such children. Other provinces have some insurance and social security plans, but we don't have this. Parents often cannot afford rehabilitative services… rehabilitation services are costly, and poor parents cannot afford them from private sectors." (Healthcare provider, Sukkur).*

The government has been working to establish health insurance coverage, but currently, there is no free insurance mechanism that directly benefits children and their well-being. Government officials have described ongoing negotiations with private insurers, but coverage has yet to be secured:

*"The government has been working on the feasibility of affordable insurance for the disabled population, and we will start free health insurance for children with disabilities soon. Currently, we are negotiating with private insurance companies, and providing health insurance coverage is a challenging task." (Government Official, Karachi).*

Inadequate budget allocation and the absence of health insurance undermine the realization of the Sindh Disability Act for children with disabilities highlighting financial barriers for child healthcare.

## 3.4 Health information systems

This sub-theme addresses the weaknesses in information systems that prevent child disability-inclusive planning and monitoring at provincial, district lor PHC level. Two main issues are lack of disability-specific indicators and reliance on manual, incomplete data collection at health facilities.

**3.4.1 Missing disability-disaggregated data.** Sindh's HIS lacks disability-specific indicators, constraining evidence-based planning and monitoring for inclusive health system strengthening. A government official explained:

*"Patient data is collected only from a clinical domain. Our information systems do not capture essential socio-demographic characteristics, disability status, or vulnerabilities, making inclusive health planning difficult." (Government Official, Hyderabad)*

Integration of disability-disaggregated data within HIS is necessary to support compliance with rights frameworks and to guide inclusive policy reforms. Most facilities still maintain manual records, excluding disability-related variables.

## 4 Service delivery

The second major theme was service delivery, which plays a central role in shaping disability-inclusive healthcare services for children with disabilities. Within this domain, three sub-themes emerged: (i) health facilities and services, (ii) health workforce, and (iii) rehabilitation and assistive technology. Each sub-theme highlights systemic barriers in infrastructure, provider preparedness, and the availability of essential medical supplies and assistive devices.

### 4.1 Health facilities and services

This sub-theme captures how the organization, accessibility, and readiness of health facilities influence the provision of disability-inclusive care for children. Key issues include gaps in newborn screening and referral pathways as well as accessibility limitations within facility infrastructure.

**4.1.1 Weak newborn screening and referral mechanisms.** Newborn screening is crucial for early detection and intervention for impairments and so is mandated under the Act. However, our findings show that most healthcare facilities lack organized screening protocols. Healthcare providers are unaware of standardized guidelines, and there is no follow-up system for referred cases. As one provider noted:

*"When a baby has an obvious physical impairment at birth, we just tell the family to see a Child Specialist. But honestly, there's no system in place to check if they actually go or get the consultation they need." (Healthcare provider, Dadu)*

Additionally, most women in rural areas seek perinatal care from private or primary health facilities, where advanced scanning and diagnostic services are unavailable. Financial barriers further restrict access, as many women cannot afford the necessary second-trimester scans recommended for screening fetal anomalies for disability prevention. A gynecologist highlighted the lack of availability and awareness of these recommended scans for identifying abnormalities during pregnancy:

*"Some scans and tests should ideally be done in the second trimester to check for problems in the fetus, but in our area these scans aren't available. Many women are also unaware of their importance because no one informs them." (Healthcare Provider, Hyderabad).*

**4.1.2 Inaccessible health facility infrastructure.** Health facilities of all levels in Sindh lack accessible infrastructure and accommodation. They often lack ramps, accessible seating, sign language interpreters, or adjustable medical equipment for children with disabilities. A disability activist explained:

*"Mothers sit on the floor, holding their disabled children for hours. Our hospitals do not have sanctioned seating arrangements or separate queue for people and children with disabilities." (Disability Activist, Karachi)*

Another disability Activist also shared the similar perspective.

*"Nobody prioritizes to provide services to children with disabilities on priority...they receive services as other children receive from hospitals and clinics". (Disability Activist, Sukkur)*

Our interviews highlighted that while some newly constructed private hospitals have begun incorporating ramps to meet accessibility requirements, these measures remain limited in scope and often inadequate for persons with disabilities. Participants emphasized that such initiatives are frequently undertaken to satisfy building approval requirements rather than to genuinely improve accessibility. As one disability activist observed:

*"Now some private health facilities make slopes instead of proper ramps to fulfill the requirements. These slopes are not accessible to children and cause major injuries if someone tries to use them." (Disability Activist, Karachi)*

Healthcare providers also pointed out the gaps in equipment and services:

*"The sign language to guide deaf people is not placed in health facilities. The hospital equipment is not adjustable for children, and this causes a lot of challenges to treat children, especially those with physical and intellectual disabilities due to lack of safety features." (Healthcare provider, Sukkur)*

## 4.2 Health workforce

This sub-theme reflects how provider knowledge, skills, and attitudes affect the quality and inclusiveness of care for children with disabilities.

**4.2.1 Limited provider awareness of disability inclusive care.** Healthcare providers often lack awareness of the needs of children with disabilities, resulting in delayed or unsuitable care. Additionally, some providers view disability-inclusive services primarily as targeted toward adults' priority. One provider stated:

*"We don't have a mechanism for adults [with disabilities], so how can we provide such services to children? First, we should think about adults, and then we can do it for children." (Healthcare provider, Dadu).*

Our interviews highlight that training gaps exist among healthcare professionals, particularly in communication, inclusive care and behavioral management skills. Many providers rely solely on attendants for patient history rather than engaging directly with children. A pediatrician shared:

*"We always communicate with the attendants of children who are deaf, non-verbal, or autistic. These children always come with someone because we don't understand them. We diagnose their conditions based solely on the history provided by the attendants." (Healthcare provider, Hyderabad).*

Another participant emphasized the need for behavioral change and capacity-building training of healthcare providers regarding inclusive care:

*"Physicians and staff often avoid attending to children who are difficult to handle. Some children with disabilities have hygiene issues or struggle to cooperate during consultations. As a result, some physicians choose not to examine them and instead refer them to another provider or delegate the prescription to an assistant." (Healthcare provider, Karachi)*

## 4.3 Rehabilitation services and assistive technology

**4.3.1 Shortage of disability-specific medical supplies.** Healthcare facilities lack essential supplies such as pediatric wheelchairs, adaptive hearing aids, rehabilitation materials, and other disability-specific equipment. Additionally, there

is no organized system to provide free corrective surgeries or medical assistance, causing families to depend on out-of-pocket payments. A healthcare provider noted:

*"We don't have AT devices for children. Parents have to buy everything themselves, which is unaffordable for many."* *(Healthcare provider, Karachi)*

A government official confirmed the lack of structured support:

*"The government has not set any mechanism to provide free surgery services and AT devices at any hospital".* *(Healthcare provider, Hyderabad)*

**4.3.2 Limited access to affordable assistive devices and technologies.** Assistive devices are essential for mobility and independence, yet their availability in public facilities for children is absent. Caregivers often rely on private suppliers, where costs and child specific affordable devices is the major concern. A disability activist stated:

*"Assistive devices are so expensive in the private market that most parents [of children with disabilities] either don't get or rely on outdated or second-hand devices."* *(Healthcare Provider, Karachi)*

A physician emphasized the challenges posed by children's growth and the need for customized devices:

*"Children are growing rapidly, and they need appliances according to their growth pattern and age. Based on the severity of disability, children need customized assistive devices. In our area, children do not get even basic devices to support their basic needs."* *(Healthcare provider, Sukkur).*

Our interviews highlighted that these children require growth related and customized assistive devices as they grow older. However, the absence of public provision, lack of awareness, and high out-of-pocket costs limit children's functional development.

## 5 Discussion

Our findings reveal that weaknesses in governance, leadership, service delivery, health financing, and information systems hinder the effective implementation of the Sindh Empowerment of Persons with Disabilities Act 2018, in Sindh province. Children with disabilities face intersectional barriers to accessing health services because these services are not prioritized to meet their needs. Applying the Missing Billion disability inclusive health system framework, we show how interconnected factors at different levels obstruct equitable healthcare access for children with disabilities, despite the mandate of the Act.

### 5.1 Governance and leadership

Although the SEPD Act has legal mandate of disability inclusive health, weak governance structures hinder its role in promoting inclusive health services across the province. Global evidence suggests that strong governance mechanism, cross-sector collaboration, and accountability are essential for disability-inclusive healthcare services [24,25]. For children with disabilities, who require coordinated, continuous, and age-appropriate care such governance failures translate into delayed identification, disrupted referral pathways, and inconsistent service provision. Despite Act provisions guaranteeing free healthcare, assistive devices, and subsidized insurance (Box 1), there are no earmarked budget lines within provincial health or social welfare allocations to support these commitments. However, similar challenges have been reported from various countries such as Chile, South Africa, Nigeria and Indonesia, where disability-inclusive health services are mandate of international and national commitments, but implementation of these policies and initiatives remained weak and poorly implemented [26–31]. Previous studies also suggest that governance and leadership structure, policy

continuity, and mechanisms of relevant stakeholder collaboration across health, social welfare, and education sectors are essential for translating disability-inclusive legislation and policies into implementation [32–34].

## 5.2 Health financing

Financial barriers continue to pose a long-standing challenge to implementation of the Act, linked to broader governance issues. While the SEPD Act (mentioned in Box 1) requires free healthcare for children with disabilities, there has been no designated budget to fulfill these obligations. The evidence suggests that children with disabilities utilize healthcare services more frequently and incurred higher out of pocket and catastrophic health expenditure than those without disabilities [35,36]. The lack of designated funds within provincial health and social welfare budgets indicates weak institutional accountability and limited prioritization of disability-inclusive health planning.

Besides budget issues, the absence of an insurance system limits access to vital health and rehabilitation services for children with disabilities. Other provinces such as: Punjab, Khyber Pakhtunkhwa, and Baluchistan have launched the Sehat Sahulat Program to provide better financial protection for children with disabilities, but such an initiatives is lacking in Sindh province [32]. Where the health insurance mechanisms are not placed, caregivers face significant out-of-pocket costs for treatment, rehabilitation, and assistive devices.

The Global Disability Inclusion Report estimates that exclusion costs LMICs up to 7% of GDP and that households with children with disabilities are disproportionately exposed to catastrophic expenditures [1]. For example, one-third of families in Iran faced catastrophic costs when a child had a disability [37]. A systematic review reported that the annual burden of childhood disability ranged $450–69,500 worldwide [38]. Some countries such as Indonesia ensure health insurance coverage for immunization, outpatient care and assistive devices [39]. Our findings highlight the importance of implementing child disability-inclusive health financing strategy with clearly earmarked budgets, tailored insurance coverage including AT devices as National Health Insurance Program [24,40–42].

## 5.3 Health information systems

The lack of disability-disaggregated data and reliance on manual, incomplete record-keeping at health facilities limits evidence-based planning at both district and provincial levels. The absence of standardized disability indicators in Sindh's HIS prevents monitoring of service utilization and unmet needs of children with disabilities. Integrating disability markers into routine health reporting, digitizing records, and developing interoperable systems are crucial steps. Evidence from LMICs shows that standardized disability data collection boosts health system accountability and aids policy improvements [36]. Similar trends are evident in high-income regions such as Europe, where despite better infrastructure, health systems still lack robust disability-disaggregated data, limiting progress towards equitable UHC [43].

## 5.4 Service delivery

Barriers in service delivery reflect systemic gaps in infrastructure, workforce readiness, and rehabilitation support. In Pakistan, more than 70 percent of primary healthcare in Pakistan is delivered through private sectors [44]. Private sector engagement is critical for the provision of disability-inclusive services that are widely accessible for caregivers of children with disabilities. Our study reported that healthcare providers have limited awareness and inadequate training to address communication and behavioral needs of children with disabilities, leading to inappropriate or delayed care in the province. Previous studies indicate that disability-inclusive training of healthcare providers to improve communication and patient-centered care significantly improves health outcomes for children with disabilities [45–47]. Rehabilitation services and assistive devices remain scarce and unaffordable, restricting children's functioning and long-term wellbeing. Addressing these challenges necessitates investment in inclusive infrastructure, integration of child disability competency into pre- and in-service training, and expansion of rehabilitation and assistive technology services [48,49]. Additionally, the

UNICEF global report on children with disabilities documents that children with disabilities face higher risks of preventable illnesses, reduced vaccination coverage, and increased under-five mortality, emphasizing the urgency of implementing inclusive newborn screening and preventive services [2].

### 5.5 Policy implications

The findings indicate that translating the SEPD Act 2018 into meaningful health gains for children with disabilities requires coordinated health system action rather than legal mandates alone. Strengthening intersectoral governance between DEPD and health authorities, accompanied by earmarked financing and disability-inclusive insurance, is essential to reduce financial and service access barriers. Integrating standardized disability indicators into routine health information systems would enable early identification, monitoring, and accountability. Concurrently, investment in accessible infrastructure, provider training, rehabilitation services, and regulated engagement of private healthcare providers is necessary to ensure that legal entitlements under the Act result in equitable and sustained healthcare for children with disabilities.

## 6 Strengthen and limitation of study findings

This study possesses several noteworthy strengths. It stands as one of the first qualitative investigations in Pakistan addressing disability-inclusive healthcare from this perspective, integrating insights from providers, policymakers, and caregivers. Furthermore, the focus is on the implementation challenges associated with the SEPD Act, analyzed through the lens of the Missing Billion Health System Framework.

However, there are some limitations. We focused on the officials and did not include interviews with children with disabilities or their caregivers, potentially missing first-hand experiences of demand-side barriers to access health services. Including their perspectives would have further clarified additional challenges from the lens of the users. There is also a lack of quantitative data to triangulate the qualitative findings in this study. Despite these limitations, this study provides valuable insights into gaps in disability-inclusive healthcare services and offers evidence-based recommendations to strengthen policy implementation in Sindh province.

## 7 Conclusions

Despite the SEPD Act's legal mandate, systemic weaknesses in governance and finance continue to constrain the realization of disability-inclusive healthcare for children with disabilities in Sindh. Weak intersectoral coordination, limited accountability, and the absence of dedicated budgets or insurance coverage for children with disabilities have hindered the translation of legislative intent into practice. Strengthening governance through clear institutional mandates, sustainable financing, and provincial accountability is essential to operationalize the SEPD Act. Equally important are reforms in service delivery and health information systems.

Future research should identify scalable models for financing and implementing child-focused disability services in Pakistan. Applying the Missing Billion Health System Framework, this study emphasizes that legislative progress must be accompanied by operationalization of policy reforms to achieve equitable health outcomes for children with disabilities.

### Supporting information

**S1 Text. Interview guide used for interviews.**
(DOCX)

### Author contributions

**Conceptualization:** Muhammad Asim.

**Data curation:** Muhammad Asim.

**Formal analysis:** Muhammad Asim.

**Funding acquisition:** Muhammad Asim.

**Investigation:** Muhammad Asim, Waqas Hameed.

**Methodology:** Muhammad Asim.

**Project administration:** Muhammad Asim, Waqas Hameed.

**Resources:** Dalia Chowdhury, Hannah Kuper.

**Supervision:** Muhammad Asim, Waqas Hameed, Hannah Kuper.

**Validation:** Sara Rotenberg, Dalia Chowdhury, Abid Lashari, Hannah Kuper.

**Writing – original draft:** Muhammad Asim.

**Writing – review & editing:** Muhammad Asim, Waqas Hameed, Sara Rotenberg, Dalia Chowdhury, Abid Lashari, Munazza Gillani, Abdul Ghaffar, Hannah Kuper.

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
