## [Decision Letter · Decision Letter 0]

15 Jan 2026

PGPH-D-25-03724

Bringing Legislation to Life: Navigating Intersectional Barriers in Implementing the Sindh Empowerment of Persons with Disabilities Act (2018) for the Healthcare of Children with Disabilities

Dear Dr. Asim,

Thank you for submitting your manuscript to PLOS Global Public Health. After careful consideration, we feel that it has merit but does not fully meet PLOS Global Public Health’s publication criteria as it currently stands. Therefore, we invite you to submit a revised version of the manuscript that addresses the points raised during the review process.

The manuscript has been assessed by two reviewers and their comments can be found below and in the attached document. Reviewer 1 has provided some clear points that need to be thoroughly addressed in the revised manuscript including justification for certain methodological decisions and improvements to the discussion sections and inclusion of potential implementation recommendations. Please review their comments and make the appropriate revisions to your manuscript.

A letter that responds to each point raised by the editor and reviewer(s). You should upload this letter as a separate file labeled ’Response to Reviewers’.A marked-up copy of your manuscript that highlights changes made to the original version. You should upload this as a separate file labeled ’Revised Manuscript with Track Changes’.An unmarked version of your revised paper without tracked changes. You should upload this as a separate file labeled ’Manuscript’.

We look forward to receiving your revised manuscript.

Kind regards,

Emma Campbell, Ph.D

Staff Editor

Journal Requirements:

i. Please clarify all sources of financial support for your study. List the grants, grant numbers, and organizations that funded your study, including funding received from your institution. Please note that suppliers of material support, including research materials, should be recognized in the Acknowledgements section rather than in the Financial Disclosure.

ii. State the initials, alongside each funding source, of each author to receive each grant. For example: “This work was supported by the National Institutes of Health (####### to AM; ###### to CJ) and the National Science Foundation (###### to AM).”

iii. State what role the funders took in the study. If the funders had no role in your study, please state: “The funders had no role in study design, data collection and analysis, decision to publish, or preparation of the manuscript.”

iv. If any authors received a salary from any of your funders, please state which authors and which funders.

3. Please ensure that your Ethics Statement is available in its entirety at the beginning of your Methods section, under a subheading ’Ethics Statement’.

4. Please upload separate figure files in .tif or .eps format. Also, remove the figures from your manuscript file but keep the legends.

5. In the online submission form, you indicated that “The data will be available upon request”.

3. Uploaded as supplementary information.

Additional Editor Comments (if provided):

Reviewers’ comments:

Reviewer’s Responses to Questions

**Comments to the Author**

1. Does this manuscript meet PLOS Global Public Health’s publication criteria? Is the manuscript technically sound, and do the data support the conclusions? The manuscript must describe methodologically and ethically rigorous research with conclusions that are appropriately drawn based on the data presented.

Reviewer #1: Yes

Reviewer #2: Yes

2. Has the statistical analysis been performed appropriately and rigorously?

Reviewer #1: Yes

Reviewer #2: Yes

3. Have the authors made all data underlying the findings in their manuscript fully available (please refer to the Data Availability Statement at the start of the manuscript PDF file)?

Reviewer #1: Yes

Reviewer #2: Yes

4. Is the manuscript presented in an intelligible fashion and written in standard English?

Reviewer #1: Yes

Reviewer #2: Yes

5. Review Comments to the Author

Reviewer #1: Thank you for the opportunity to review this manuscript. I have answered YES to the relevant questions above, as I believe this study addresses an important, timely, and policy-relevant topic and has the potential to make a meaningful contribution to the literature on disability-inclusive health systems in low- and middle-income countries.

The manuscript is well aligned with the scope of PLOS Global Public Health, particularly in its focus on health equity, implementation gaps, and the application of a health systems framework. The use of the Missing Billion Disability-Inclusive Health System Framework, the qualitative exploratory design, and the inclusion of diverse stakeholders (policymakers, health administrators, clinicians, and disability advocates) are notable strengths.

That said, I recommend major revisions to strengthen clarity, rigor, and alignment between the stated objectives and the presented findings. In particular, the authors are encouraged to address the following key areas:

Clarify the child-specific focus:

While the manuscript emphasizes children with disabilities, many findings and quotations refer broadly to persons with disabilities. The authors should more explicitly distinguish child-specific barriers (e.g., newborn screening, pediatric rehabilitation, growth-related assistive technology, caregiver dependence) or clearly explain how child-relevant implications were derived from broader system-level narratives.

Enhance methodological transparency:

The data analysis process would benefit from clearer description, particularly regarding the combined inductive–deductive approach and the role of the Missing Billion framework in coding versus interpretation. Additional justification for the use of Microsoft Excel for qualitative analysis should also be provided.

Address data availability requirements:

The current data availability statement (“data available upon request”) does not fully align with PLOS data-sharing policies. The authors should either provide de-identified qualitative data in an appropriate repository or clearly justify ethical or legal restrictions, specifying what data can be shared and under what conditions.

Improve language, consistency, and use of disability-sensitive terminology:

The manuscript contains multiple grammatical issues and instances of non–person-first language (e.g., “not disable”). A thorough language edit is strongly recommended, along with consistent use of disability-inclusive terminology and formatting throughout.

Strengthen the discussion and policy implications:

The discussion section is comprehensive but lengthy and somewhat repetitive. It would benefit from tighter synthesis, clearer linkage between findings and specific provisions of the Sindh Empowerment of Persons with Disabilities Act (2018), and more concise, actionable policy recommendations.

I did not identify concerns related to dual publication, plagiarism, or major breaches of research or publication ethics. Ethical approval is clearly stated, and reflexivity is appropriately acknowledged, though this section could be strengthened with more critical reflection on positionality and power dynamics.

Overall, this is a strong and promising manuscript. With careful revision addressing the points above, it has good potential for publication in PLOS Global Public Health and for informing disability-inclusive health policy and practice in Pakistan and similar contexts.

Reviewer #2: This is a much needed study to highlight the gap between legislative aspirations which are laudable but are not matched in the resourcing, coordination, delivery and evaluation of services which would have a major impact on health inequalities affecting a vulnerable segment of the population namely disabled children. The Missing Billion Health System Framework is well suited to structure the analysis required in this study

6. PLOS authors have the option to publish the peer review history of their article (what does this mean?). If published, this will include your full peer review and any attached files.

**Do you want your identity to be public for this peer review?** For information about this choice, including consent withdrawal, please see our Privacy Policy.

Reviewer #1: **Yes:** WARUKIRA JULIUS NJUGUNA

Reviewer #2: No

Figure Resubmissions:

After uploading your figures to PLOS’s NAAS tool - https://ngplosjournals.pagemajik.ai/artanalysis, NAAS will process the files provided and display the results in the “Uploaded Files” section of the page as the processing is complete. If the uploaded figures meet our requirements (or NAAS is able to fix the files to meet our requirements), the figure will be marked as “fixed” above. If NAAS is unable to fix the files, a red “failed” label will appear above. When NAAS has confirmed that the figure files meet our requirements, please download the file via the download option, and include these NAAS processed figure files when submitting your revised manuscript.

---

## [Decision Letter · Decision Letter 1]

13 May 2026

Bringing Legislation to Life: Navigating Intersectional Barriers in Implementing the Sindh Empowerment of Persons with Disabilities Act (2018) for the Healthcare of Children with Disabilities

PGPH-D-25-03724R1

Dear Dr Asim,

We are pleased to inform you that your manuscript ’Bringing Legislation to Life: Navigating Intersectional Barriers in Implementing the Sindh Empowerment of Persons with Disabilities Act (2018) for the Healthcare of Children with Disabilities’ has been provisionally accepted for publication in PLOS Global Public Health.

If your institution or institutions have a press office, please notify them about your upcoming paper to help maximize its impact. If they’ll be preparing press materials, please inform our press team as soon as possible -- no later than 48 hours after receiving the formal acceptance. Your manuscript will remain under strict press embargo until 2 pm Eastern Time on the date of publication. For more information, please contact globalpubhealth@plos.org.

Best regards,

Somayeh Hessam

Academic Editor

Reviewer Comments (if any, and for reference):

Reviewer’s Responses to Questions

**Comments to the Author**

1. If the authors have adequately addressed your comments raised in a previous round of review and you feel that this manuscript is now acceptable for publication, you may indicate that here to bypass the “Comments to the Author” section, enter your conflict of interest statement in the “Confidential to Editor” section, and submit your “Accept” recommendation.

Reviewer #1: All comments have been addressed

Reviewer #2: All comments have been addressed

2. Does this manuscript meet PLOS Global Public Health’s publication criteria? Is the manuscript technically sound, and do the data support the conclusions? The manuscript must describe methodologically and ethically rigorous research with conclusions that are appropriately drawn based on the data presented.

Reviewer #1: Yes

Reviewer #2: Yes

3. Has the statistical analysis been performed appropriately and rigorously?

Reviewer #1: Yes

Reviewer #2: Yes

4. Have the authors made all data underlying the findings in their manuscript fully available (please refer to the Data Availability Statement at the start of the manuscript PDF file)?

Reviewer #1: Yes

Reviewer #2: Yes

5. Is the manuscript presented in an intelligible fashion and written in standard English?

Reviewer #1: Yes

Reviewer #2: Yes

6. Review Comments to the Author

**Reviewer #1:** Thank you for the opportunity to review this manuscript. After a careful and thorough assessment, I find the study to be well-structured, methodologically sound, and highly relevant to the field of disability-inclusive health systems.

The manuscript clearly articulates the challenges in implementing the Sindh Empowerment of Persons with Disabilities Act (2018), particularly in relation to healthcare access for children with disabilities. The use of a qualitative exploratory design, along with the application of the Missing Billion framework, strengthens the analytical depth and provides meaningful insights for policy and practice.

The findings are presented in a coherent and logical manner, and the discussion effectively links the results to broader global and regional contexts. Importantly, the study contributes valuable evidence that can inform policymakers, practitioners, and researchers working in disability inclusion and public health.

I did not identify any major concerns regarding research ethics, dual publication, or methodological integrity. The manuscript meets the standards for academic publication, and I believe it is suitable for publication in its current form.

Minor editorial refinements may further enhance clarity, but these are not substantial and do not affect the overall quality of the work.

Overall, this is a strong and impactful contribution to the literature.

**Reviewer #2:** the authors have received detailed comments from a number of reviewers. They have provided systematic responses to these and have altered the manuscript accordingly

7. PLOS authors have the option to publish the peer review history of their article (what does this mean?). If published, this will include your full peer review and any attached files.

**Do you want your identity to be public for this peer review?** For information about this choice, including consent withdrawal, please see our Privacy Policy.

Reviewer #1: **Yes:** WARUKIRA JULIUS NJUGUNA

Reviewer #2: No
